# Structure and Function of Human Matrix Metalloproteinases

**DOI:** 10.3390/cells9051076

**Published:** 2020-04-26

**Authors:** Helena Laronha, Jorge Caldeira

**Affiliations:** 1Centro de investigação interdisciplinar Egas Moniz, Instituto Universitário Egas Moniz, 2829 Caparica, Portugal; h.laronha@campus.fct.unl.pt; 2UCIBIO and LAQV Requimte Faculdade de Ciências e Tecnologia, Universidade Nova de Lisboa, 2829-516 Caparica, Portugal

**Keywords:** matrix metalloproteinases, TIMP, collagen

## Abstract

The extracellular matrix (ECM) is a macromolecules network, in which the most abundant molecule is collagen. This protein in triple helical conformation is highly resistant to proteinases degradation, the only enzymes capable of degrading the collagen are matrix metalloproteinases (MMPs). This resistance and maintenance of collagen, and consequently of ECM, is involved in several biological processes and it must be strictly regulated by endogenous inhibitors (TIMPs). The deregulation of MMPs activity leads to development of numerous diseases. This review shows MMPs complexity.

## 1. Extracellular Matrix—Collagen

The extracellular matrix (ECM) is a macromolecules network, composed of collagen, enzymes and proteins (Figure 1), that promote a structural and biochemical support.

The ECM has many components [1]: Fibers (collagen, elastin, laminin, and fibronectin), proteoglycans (syndecan-1 and aggrecan), glycoproteins (tenascin, vitronectin and entactin) and polysaccharides (hyaluronic acid) [2], that regulate cell migration, growth, and differentiation [3].

The collagen is the most abundant protein in ECM which gives structural support for cells [1,3]. Depending of mineralization degree, the tissues can be divided in rigid (bones) or compliant (sinews) or have a gradient between these two states (cartilage) [4]. Collagen can come in two main forms: fibrillar (type I, II, III, V, and XI) and non-fibrillar, the latter includes facit-fibril associated collagens with interrupted triple helix (type IX, XII, XIV, XIX, and XXI); short chain (type VIII and X); basement membrane (type IV); multiplexin (type XV and XVIII); MACIT- membrane associated collagens with interrupted triple-helix (type XIII and XVII) and others types (Type VI, VII, and VIII). The collagen protein consists of three α chains, in triple helix, where two chains are chemically similar (α_1_ and α_2_), with approximate dimensions of 300 × 1.5 nm [3,5]. The triple helix is divided into five d-segments with D1–D4 having a length of 67 nm and D5 equal to 0.46 nm [3,5]. The high glycine content is important for collagen helix stabilization as it allows collagen fibers to combine, facilitating hydrogen bridges and cross-link formation [5].

The collagen synthesis involves several steps [5] (Figure 2). The mRNA is transcripted by ribosome, forming pre-propeptide. The following three stages occur in the endoplasmic reticulum: (1) The *N*-terminal signal sequence is removed; (2) hydroxylation of lysine and proline by prolyl hydroxylase and lysyl hydroxylases takes place, together with (3) glycosylation of lysine, forming the pro-collagen. This structure is a triple helix chain, but with the unwound terminals. The removal of these terminals occurs in extracellular medium, by collagen peptidases, forming tropocollagen. The microfibril collagen is formed by lysyl oxidase that packs together five tropocollagen chain [3], which have a characteristic image. At intervals of 67 nm, one zone has the roll of all tropocollagen and another zone has one less—“gap”. The fibril collagen is composed of various microfibril collagen group and the alternating overlap and gap regions create the characteristic “bright and dark” d-banding pattern [3].

Its triple helix conformation makes collagen resistant to many proteases. The enzymes able to cleave this structure are capthesin K and enzymes with collagenolytic activity (MMPs-1, -2, -8, -13, -14, and -18) [3,4]. The collagen type I, II and III have a specific cleavage sequence: (Gln/Leu)-Gly#(Ile/Leu)-(Ala/Pro), which is located at 3/4 of *N*-terminal and this is crucial for collagen degradation [6,7] (Figure 3). The denatured collagen is called gelatin, which can be further degraded by gelatinases [4,6]. The collagen degradation is involved in many biological processes, such as embryogenesis, morphogenesis, tissue remodulation, angiogenesis, and wound healing [4].

The ECM degradation is also an important process in development, morphogenesis, tissue repair and remodulation [8], and can affect the cellular behavior and phenotype [6,9]. The excessive degradation of ECM leads to metabolic and immune diseases, cancer and cardiovascular disorders (hypertension, atherosclerosis and aneurysm) [2,3,6].

## 2. Metzincs Superfamily—Matrix Metalloproteinases (MMPs)

The metzincs are a family of multidomain zinc-dependent endopeptidases, where the metalloproteinases such as matrix metalloproteinases (MMPs) [10], the desintegrins and metalloproteins (ADAMS), the ADAMs with thrombospondin motif (ADAMTS), were snapalysins, leishmanolysins, pappalysins belong [1,6,8,11,12,13,14]. All of them have a common conserved domain: HExxHxxGxxH [6,11,12,13,15,16,17,18,19] and, in case for MMPs, this sequence is HEBGHxLGLxHSBMxP [13].

The MMPs were first described in 1949 [20], as depolymerizing enzymes that facilitate tumor growth, by making connective tissue stroma, including small blood vessels that are more fluid [16,20,21]. In 1962, a MMP collagenase was isolated and characterized as an enzyme responsible for tadpole tail resorption [1,2,6,13,14,16,21,22], by Gross and Lapiere [22]. Over the next 20 years, several enzymes were purified, but it was in 1985 that the area has developed more significantly [16,21]. Taken together it has been demonstrated that MMPs are present in virus, archeabacteria, bacteria, plants, nematodes, and animals [6,13].

The MMPs are a family of proteolytic enzyme that have different substrates, but which share similar structural characteristics [1,2,11,13,23]. The active site is zinc-dependent [1,2,6,11,13,15,16,17,18,24,25] and is highly conserved, with three histidine residues bound to catalytic zinc [6,11,12,15] (Figure 4). They are functional at neutral pH [9,18].

MMPs have been recently recognized as biomarkers in several fields (diagnosis, monitoring, and treatment efficacy) [19], since their overexpression in diseases conditions is specific and elevated [19]. Huang et al. [26] related that MMP-9 represents a potential biomarker which is overexpressed in several types of tumors (colarectal carcinoma, breast, pancreatic, ovaria, cervical, osteosarcoma non-small cell lung cancer (NSCLC), and giant cell tumor of bone (GCTB)), which makes MMP-9 a preferential candidate for the early detection of these diseases [26]. The elevated levels of MMP-9 can be detected in plasma or blood, show triple protein levels compared to healthy patients [19,27,28]. The MMP-9 in contrast with other proposed biomarkers (MMP-1 and -3), also presents increased levels in specific risk group to atherosclerosis [19], as demonstrated by Rohde et al. [29]. Nilsson et al. [30] studied the MMPs-1, -3, -7, -10, and -12 in plasma and demonstrated that MMP-7 and -12 were elevated in type 2 diabetes, which is related to atherosclerosis and coronary events [30].

## 3. MMPs Functions

The principal biologic function of MMPs is degradation of ECM proteins and glycoproteins [1,2,6,8,11,12,14,16,18,19,24,31], membrane receptors [13,14,24,31], cytokines [11,13,14,31], and growth factors [11,13,14,24,31]. The MMPs are involved in many biologic processes, such as, tissue repair and remodulation [1,2,9,13,14,16,18,24], cellular differentiation [1,2,18], embryogenesis [2,6,9,13,14], morphogenesis [2,24], cell mobility [9,18], angiogenesis [1,2,6,9,14,18,24], cell proliferation [1,2], and migration [1,2], wound healing [1,2,6,9,13,14,15,18], apoptosis [1,2,18], and main reproductive events such as ovulation [13,14] and endometrial proliferation [18].

The deregulation of MMP activity leads to the progression of various pathologies, that can be grouped into [1,6,14,18,19,24,25]: (1) Tissue destruction, (2) fibrosis, and (3) matrix weakening. The overexpression of MMPs is involves in several diseases (Table 1).

The MMPs are involved in the development and progression of atherosclerosis, which is correlated with cardiovascular diseases [19]. Their activity promotes the loss of collagen, elastin and other ECM proteins, inducing the necrotic core of atherosclerotic plaque that leads to myocardial infarction or a stroke [19].

Some studies show intracellular localization of MMP-2 in cardiac myocytes and colocalization of MMP-2 with troponin I in cardiac myofilaments [32]. The MMP-2 activity has also been detected in nuclear extracts from human heart and rat liver [32]. Poly ADP-ribose polymerase is a nuclear matrix enzyme involved in DNA repair and is susceptible to cleavage by MMP-2, in vitro its cleavage being blocked by MMPs inhibitors [32]. The presence of MMP-2 in nucleus could play a role in poly ADP-ribose polymerase degradation and affect DNA repair [32].

## 4. Types of MMPs

In vertebrates, there are 28 different types of MMPs [1,2,8,9,11,12,13,16,17], at least 23 are expressed in human tissue [1]. MMPs can be subdivided according to bioinformatic analysis, in 5 types [23]:Non-furin regulated MMPs (MMP-1, -3, -7, -8, -10, -12, -13, -20, and -27);MMPs bearing three fibronectin-like inserts in the catalytic domain (MMP-2 and -9);MMPs anchored to the cellular membrane by a *C*-terminal glycosylphosphatidylinositol (GPI) moiety (MMP-11, -17 and -25);MMPs bearing a transmembrane domain (MMP-14, -15, -16, and -24) andAll the other MMPs (MMP-19, -21, -23, -26 and -28).

But, MMPs can be also subdivided according to substrate specificity, sequential similarity and domain organization (Appendix A) into: Collagenases, gelatinases, stromelysins, metrilysins, membrane-type MMPs, and other MMPs [1,2,8,9,12,13,14,16,17,18,19,25].

Collagenases (Appendix A, Table A1) cleave some ECM proteins and other soluble proteins, but the most important role of this type of MMPs is the cleavage of fibrillar collagen type I, II, III, IV and XI into two characteristic fragments, 1/4 *C*-terminal and 3/4 *N* -terminal [1,6,8,9,12,14,17,21]. This process takes place in two steps: first MMP unwind triple helical collagen and then hydrolyze the peptide bonds [1]. The hemopexin domain is essential for cleaving native fibrillar collagen while the catalytic domain can cleave non-collagen substrates [1].

Gelatinases (Appendix A, Table A2) play an important role in many physiological processes, such as ECM degradation and remodeling, osteogenesis and wound healing [9]. Gelatinases degrade gelatin [12,17], collagen type IV [8,17,21], V [8,17], VIII, X, XI [8,17], and XIV, elastin [21], proteoglycan core proteins [8,17], fibronectin [21], laminin [17,21], fibrilin-1, and TNF-α and IL-1b precursor [9], due to the existence of three repeated fibronectin type II domain [1,8,12,17,18], which binds gelatin [1], collagen, and laminin [1]. MMP-2 is primarily a gelatinases, but can acts like collagenase, albeit in a weaker manner [1]. MMP-2 degrades collagen in two steps: first by inducing a weak interstitial collagenase-like collagen degradation and then by promoting gelatinolysis using the fibronectin-like domain [1]. MMP-9 can act as collagenases and gelatinase [1].

Gelatinases are involved in physiological and pathological states, such as, embryonic growth and development, angiogenesis, vascular diseases, inflammatory, infective diseases, degenerative diseases of the brain and tumor progression [1]. Tumor metastasis is a process that involves the release of tumor cells, their migration through blood vessels, penetration into the blood and lymphatic system and their adhesion into the endothelial vessel and extravasation into tissue [11]. The activity of gelatinases is crucial for metastatic cell output and metastasis site entry [11]. Increased expression and activity of gelatinases have been described in malignant diseases such as breast, urogenital, brain, lung, skin and colorectal cancer [11].

Stromelysines (Appendix A, Table A3) have the same domain arrangement as collagenases, but do not cleave interstitial collagen [1]. MMP-3 and -10 are closely related by their structure and substrate specificity [1,8,9,17], while MMP-11 is distantly related [1]. The intracellular activation of MMP-1 is regulated by 10 amino acids insert, localized between the pro- and catalytic domains (RXRXKR), which is recognition by Golgi-associated proteinase furin.

The main characteristic of the matrilysins (Appendix A, Table A4) is the lack of hemopexin domain, present in the other MMPs [9,12,17,18]. This MMP group has a specific feature in the amino acid sequence with a threonine residue adjacent to the Zn^2+^- binding site [1].

Membrane-type metalloproteinases (MT-MMP; Appendix A, Table A5) contain a furin-like pro-protein convertase recognition site (RX[R/K]R) in their pro-domain *C*-terminal [1,8,17,18], allowing pro-enzyme activation by proteolytic removal of this domain. They are activated intracellularly and the active enzymes are expressed on the cell surface [1]. This group can be subdivided into: type I transmembrane proteins (MMPs-14, -15, -16, and -24) [1,8,9,17,18] and glycosylphosphatidylinositol (GPI) anchored proteins (MMPs-17 and -25) [1,8,9,17,18]. The type I transmembrane protein have about 20 amino acids long cytoplasmic tail following the transmembrane domain [1]. MT-MMPs have the insert of eight amino acids in the catalytic domain, which in case of MMP-14 consists of PYAYIREG and this sequence can influence on conformation of the active site cleft [1].

## 5. Structure

Lovejoy et al. reported the first structure of MMP-inhibitor complex [33]. This structure reveals that the active site of MMP is a deep cavity and moreover that the catalytic domains of MMPs share a sequential similarity, where the percentage of similarity ranges between 33% (between MMP-21 and MMP23) and 86% (between MMP-3 and MMP-10) [33]. 3D structures of the catalytic domains of MMP-1 and -8 as well as structures of pro-MMP-3 and MMP-1 followed [33].

The most common structural features are (Figure 5 and Table 2) [1,2,8,9,13,14,15,16,17,18,19,23]:1-A signal *N*-terminal peptide with variable length, that targets the peptide for secretion;2-A pro-domain (with about 80 aa), which keeps MMP inactive and is removed when the enzyme is proteolytically activated;3-A catalytic domain (with about 160 aa), with a zinc ion, that consists of five β-sheets, three α-helixes and three calcium ions;4-A linker of variable length (14–69 aa), which links the catalytic domain to hemopexin-like domain—“hinge region”;5-A hemopexin-like domain (with about 210 aa) that is characterized by four β-propeller and6-An additional transmembrane domain with the small cytoplasmatic *C*-terminal domain, only present in MMPs-14, -15, -16 and -24.

All MMPs are synthesized with an *N*-terminal signal sequence, which is removed in the endoplasmic reticulum, producing pro-enzymes [13].

The pro-domain (Figure 6) contains three α-helixes [8,12,13,17,18]. In the case of MMP-1 and -2 [13], the first loop has a cleavage region—“bait-region” [8,17]—which is protease sensitive [12,13,18] (EKRRN for pro-MMP-1 and SCN*LF for pro-MMP-2) [13]. The α(3)-helixes are followed by a “cysteine switch” [2,6,12,18], a very conserved region (PRCGXPD) [1,2,17,18,19,21], where the sulfhydryl group (Cys residue) coordinates the catalytic zinc [1,2,6,8,17,18,21], forming a tetrahedral coordination sphere, thereby blocking enzyme [2,15,17,18,21].

Active site (Figure 7) consists of two regions: a cavity on the surface of the protein that houses the zinc ion in the center and a specific site S1′ [16]. The catalytic domains of the known MMP share the same structural organization: Three α-helixes, five β-sheets (four parallel: β2–β1–β3–β5 and one anti-parallel: β4), connected by eight loops [8,9,12,13,16,17,19,21]. They are sphere shaped, with a diameter of ~40 Å [14]. Catalytic domain is highly conserved and in addition to catalytic zinc ions contain another Zn^2+^, that has a structural function [1,6,17,19,21], three calcium ions [6,8,9,12,15,17,19,23] and three histidine residue [1,18,19,21] that are part of a highly conserved sequence: VAAHEXGHXXGXXH [1,19,21]. The Ω-loop is found between α2-helix and α3-helix, its length and amino acid composition vary significantly among MMPs [19]. This loop is the least conserved fragment within the catalytic domain, which is most likely responsible for different selectivity of MMPs [19]. In the terminal zone of the catalytic domain, located 8 residues down from the Zn^2+^ [1], there is a region that forms the outer wall of pocket S1′ [1,19]—“met-turn” (XBMX) [1,13]. The catalytic zinc ion and the “met-turn” are conserved and also are present in ADAM and ADAMTS families, astacins and bacterial serralysins [8]. The second zinc ion and the three calcium ions, that are present is all MMPs, have only a structural function and they been crucial for maintaining protein conformation [23]. Glutamic acid adjacent to the first histidine residue is essential for the catalytic process [1,12,16,21].

Catalytic and hemopexin-like domains are linked by a proline-rich linker [9,13,17] of variable length [18] that allows inter-domain flexibility [17], enzyme stability [13] and is involved in the hydrolysis of some structurally complex substrates [9,13,23] (Figure 8). Mutations in the linker region in MMP-1 [34] and MMP-8 [35] reduce collagenolytic activity, supporting the hypothesis that correct movement and rearrangement between the catalytic domain and the hemopexin-like domain is important for activity [17].

The hemopexin-like domain has four β-propeller structural elements [8,12,13,17]. This domain is necessary for collagen triple helix degradation [8,11,17] and is important for substrate specificity [11,12,13,21] (Figure 9). It also may be essential for the recognition and subsequent catalytic degradation of fibrillar collagen, whereas the catalytic domain is sufficient for the degradation of non-collagen substrates [1]. The only MMPs that don’t have the hemopexin-like domain are MMP-7, -23, and -26 [8,15,17]. In the case MMP-23, this domain is substituted by an immunoglobulin-like domain and a cysteine-rich domain [8,15,17], located immediately after the *C*-terminal of the catalytic domain [17].

### S_1_′ Pocket Selectivity

MMPs possess six pockets (S_1_, S_2_, S_3_, S_1_′, S_2_′, and S_3_′) [19] and the fragments of the substances and inhibitors are consequently named after the pocket that they interact with (P_1_, P_2_, P_3_, P_1_′, P_2_′, and P_3_′) [2,6]. The S_1_, S_2_, and S_3_ pockets are unprimed pockets, which are localized on the right side of zinc ion [19]. The S_1_′, S_2_′, and S_3_′ pockets are the primed pockets, which are localized on the left side of zinc ion [2,19]. It has been demonstrated that the S_1_′ is the most variable pocket in MMPs [1,6,8,15,16], followed by S_2_, S_3_′, S_1_, and S_3_ pockets with an equivalent degree of variance, while the S_2_′ has the lowest variability. The S_2_′ and S_3_′ pockets are shallower than S_1_′ and are more exposed to solvent [1]. The S_3_ pocket may also contribute to substrate specificity [1].

S_1_′ pocket is the most important since it is a determining factor for substrate specificity [1,2,8,16], Its cavity [1,19], formed by the Ω-loop [19] is a highly hydrophobic. By analyzing the depth of the different S_1_′ subsites, MMPs can be divided into three different subgroups [1,2,17,19,36] (Figure 10): the shallow, the intermedium and the deep pockets. This differentiation is related to the size of amino acids: MMP-1 and -7 have an Arg^214^ and Tyr^214^ residue, respectively [13,36], instead of a leucine present in MMP-2, -3, -8, -9, -10, -12, -13, and -14 [16,36], giving origin to shallow S_1_′ pocket [15].

## 6. MMP Activity Regulation

MMP expression is closely controlled for transcription, secretion, activation, and inhibition of the activated enzyme [1,2,9,14,18]. Some MMPs are not constitutively expressed by cells, in vivo, instead their expression is induced by exogenous signals such as cytokines [1,14,16], growth factors [1,14,16], hormones [1,16], and changes in cell–matrix and cell–cell interactions [8,9,15].

MMPs are produced by various tissues and cells [1]. MMPs are secreted by connective tissue, pro-inflammatory and uteroplacental cells, such as fibroblast, endothelial cells, osteoblasts, vascular smooth muscle, macrophages, lymphocytes, cytotrophoblasts, and neutrophils [1]. MMPs are synthesized as pre-proMMPs, from which the signal peptide is removed during translation to generate proMMPs or zymogens [1]. MMPs are secreted in the form of latent precursors- “zymogens”, which are proteolytically activated in extracellular medium [1,2,6,13,14,16,17,18,19,24]. The important activation mechanism is the proteolytic removal of the pro-domain by endopeptidases, such as, other MMPs, serine proteases, plasmin or furin [1,11,12,13,14,15,16,21]. The first proteolytic attack occurs in the “bait region” [8,19,24] and the specificity of cleavage is dictated by a sequence found in each MMP [9,12,18]. This leads to the removal of part of the pro-domain, destabilizing the rest of the pro-domain [13], including cysteine switch-zinc interaction [17,24], which allows intramolecular processing [8,9,12,18]—“stepwise activation” [8,18] (Figure 11). One question that remains unanswered is how a simple bait-region cleavage can result in destabilization of the pro-domain and cause conformational changes in the zone between the pro-domain and the catalytic domain [13].

Pro-MMP-2 forms a complex with TIMP-2 through interactions of the hemopexin-like domain with the non-inhibitory *C*-terminal domain of TIMP-2 [1,2,8,17,18,25]. Formation of this complex is essential for pro-MMP-2 activation by MT1-MMP [8,17,18]. The complex then reaches the cell surface where it binds to the active site of MT1-MMP [1], via the free inhibitory *N*-terminal of TIMP-2 [2], orienting the pro-MMP-2 pro-domain adjacent to MT1-MMP [8,17,18]. With the aid of a second MT1-MMP [2], interactions between the MT1-MMP occur through their hemopexin-like domains, forming a quaternary tetrameric complex [1,17]: an MT1-MMP acts as a receptor for the pro-MMP-2-TIMP-2 complex and the other as a pro-MMP-2 activator [2,8,18]. Excess of TIMP-2 prevents this activation process by inhibiting the second MT1-MMP [2,8].

The MMP activation can also occur by physiochemical agents, such as heat, low pH, chaotropic agents and thiol-modifying agents (4-aminophenylmercurin acetate, mercury chloride and *N*-ethylmaleimide) [1], which cause disruption of the cysteine-Zn^2+^ coordination [1].

Pro-enzymes and active forms of MMP are controlled by the specific stereochemical binding of metalloproteinase-specific inhibitors (TIMPs: -1, -2, -3, and -4) [1,2,6,9,13,14,16,18,21,25] and by non-specific proteinase inhibitors such as the α1-proteinase inhibitor and α2-macroglobulin [8,9,13,14,16,17,18,21,25].

## 7. Catalytic Mechanism

Over the past 30 years, the detailed structural characterization of different MMPs allowed disentangling of the individual catalytic steps that occur at the active site during proteolysis [23].

The catalytic activity requires catalytic zinc ion and a water molecule flanked by three conserved histidine and a conserved glutamate, with a conserved methionine acting as a hydrophobic base to support the structure surrounding the catalytic zinc ion [1,2,14] (Figure 12). In the initial transition states of the MMP-substrate interaction, Zn^2+^ is penta-coordinated with a substrate’s carbonyl oxygen atom, one oxygen atom from the glutamate-bound water and the three conserved histidine residues [1]. Hydrolysis of the peptide bond begins with the nucleophilic attack of the water coordinated with zinc to the carbonyl carbon of the substrate [13,14,17,18], subsequently proton transfer to the amine nitrogen occurs through the glutamic acid residue [2,8,12,13,15,16,18,21], promoting a gem-diol reaction intermediate [9,11,23] with a tetrahedral geometry [1,2,13]. This results in the breakdown of the substrate and the release of a water molecule [1]. The peptide is stabilized at the active site, by interaction between *N*-terminal residues and S_1_′ pocket [1,12], and by new hydrogen bonds formed between *N*-terminal, glutamate and water [2,9,11,13,15,16,18,23]. The two key steps in the catalytic process involve a structural rearrangement of the active site and the fate of the two obtained peptides [23]. Particularly relevant to the catalytic mechanism is the flexibility of the loop that forms the exterior of the S_1_′ pocket [23]. The internal flexibility of the catalytic domain plays an important role for enzymatic activity, but it is also the cause of drawbacks related to inhibitor selectivity.

## 8. Conclusions

MMPs are the most important enzymes for ECM maintenance. They are also involved in several biological and pathological diseases. However, MMPs remain a challenge for science, due to their high complexity, both in terms of regulation and activity. To that end, understanding of their role in the development of diseases and their mode of action require further studies. The development synthetic inhibitors in particular, critically depend on the full understanding of the structural details about S_1_′ pocket in MMPs and its interaction with the substrate.

## Figures and Tables

**Figure 1 cells-09-01076-f001:**
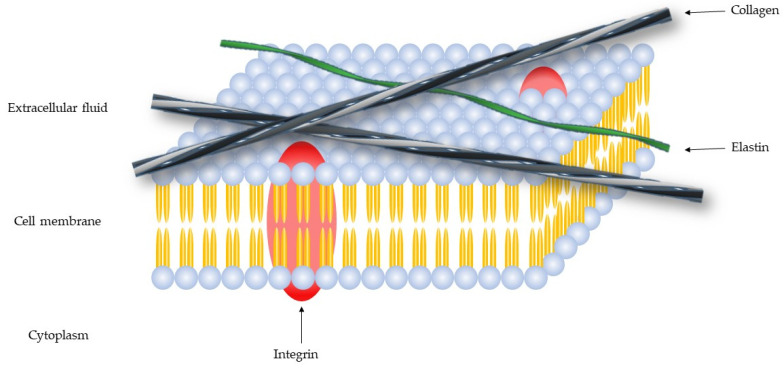
Schematic representation of the cell membrane.

**Figure 2 cells-09-01076-f002:**
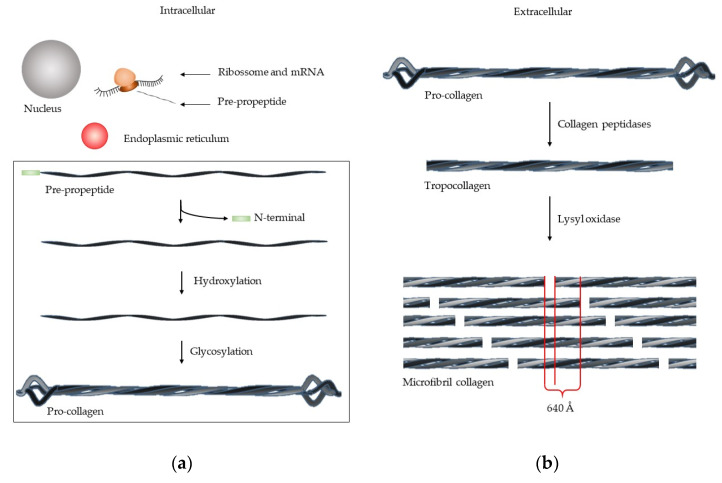
Synthesis of microfibril collagen. (**a**) In intracellular medium, the mRNA is transcripted by ribosome, forming the pre-peptide, which is then processed in endoplasmic reticulum, forming pro-collagen. (**b**) In extracellular medium, the pro-collagen is processed by collagen peptidase, forming tropocollagen. For microfibril collagen formation, the tropocollagen is processed by lysil oxidase.

**Figure 3 cells-09-01076-f003:**
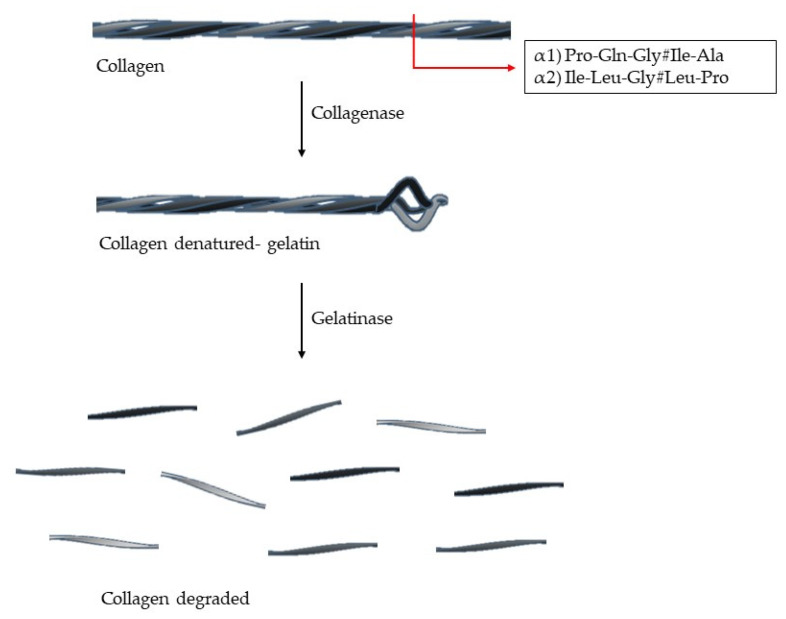
Collagen degradation. The enzyme with collagenolytic activity (collagenases) cleaves the triple helix at two fragments: 3/4 *N*-terminal and 1/4 *C*-terminal. Each chain (α1 and α2) has a specific cleavage sequence (# represents the cleavage site).

**Figure 4 cells-09-01076-f004:**
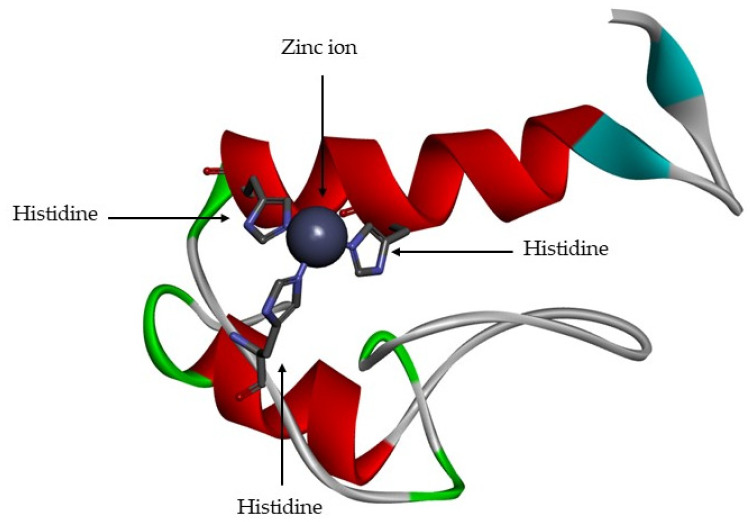
Active site of matrix metalloproteinase (MMP)-1. The zinc catalytic is represented by grey ball and the three histidine residues are represented by sticks.

**Figure 5 cells-09-01076-f005:**
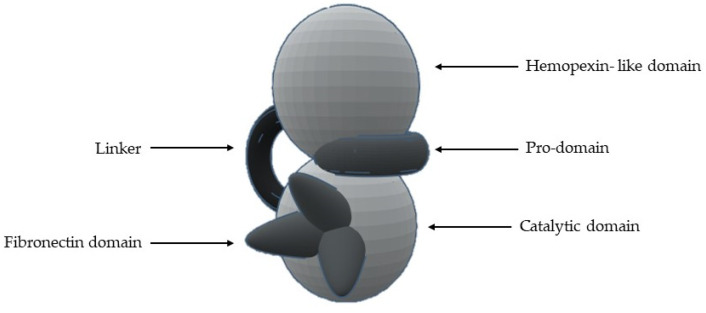
Schematic representation of the general structure of MMP.

**Figure 6 cells-09-01076-f006:**
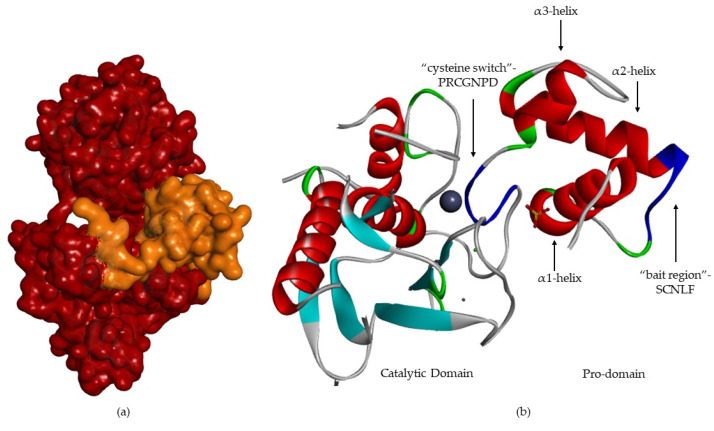
MMP-2 with pro-domain. (**a**) Surface of MMP-2, where the pro-domain is represented in orange. (**b**) Three-dimensional structure of MMP-2, where “bait-region” and “cysteine switch” are represented in blue.

**Figure 7 cells-09-01076-f007:**
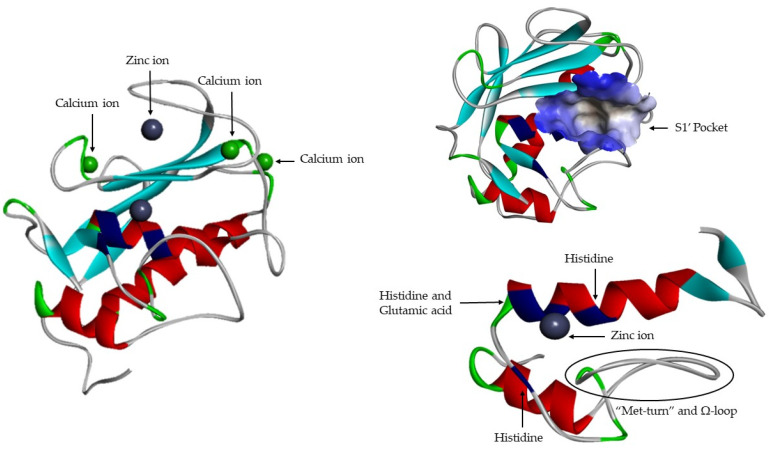
MMP-1 catalytic domain.

**Figure 8 cells-09-01076-f008:**
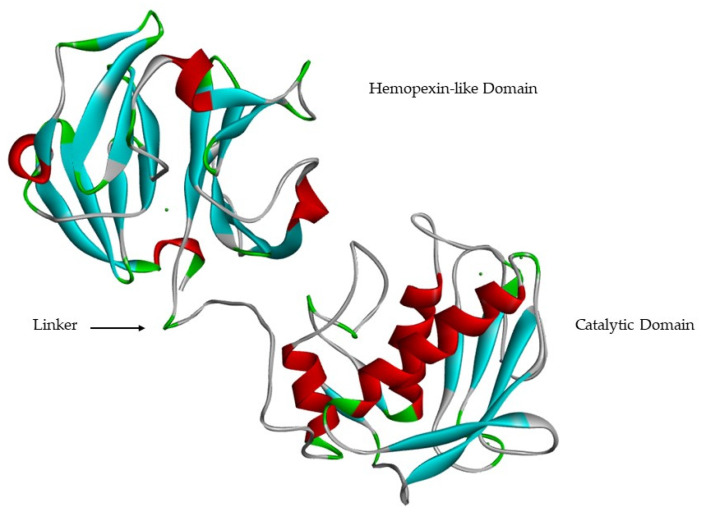
MMP-1 catalytic domain, hemopexin-like domain and linker.

**Figure 9 cells-09-01076-f009:**
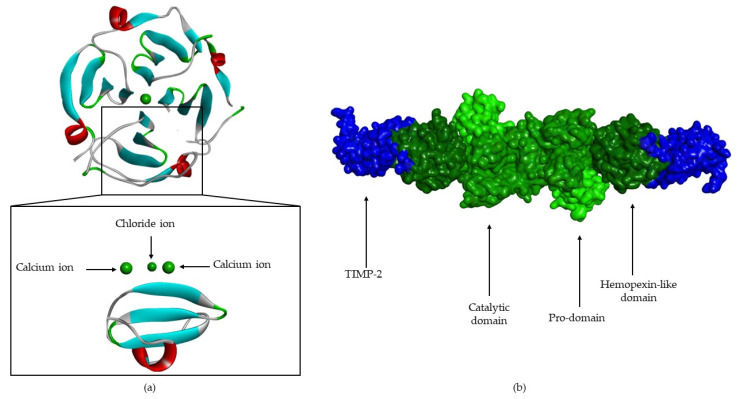
(**a**) MMP-1 hemopexin-like domain, composed to 4 β-propeller (4 β- sheets and 1 α-helix). (**b**) Surface of pro-MMP-2-TIMP-2 complex (pro-domain, catalytic domain and hemopexin-like domain are represented in green; TIMP-2 is represented in blue).

**Figure 10 cells-09-01076-f010:**
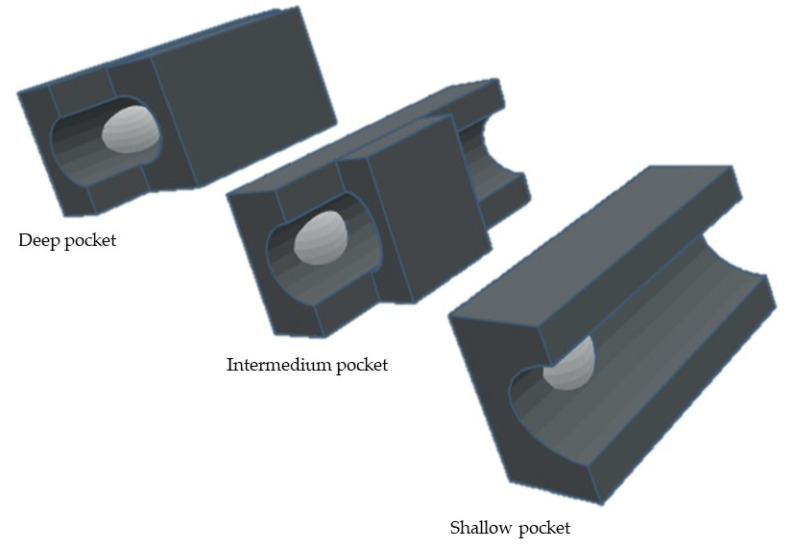
Schematic representation of different subgroups of S_1_′ pocket.

**Figure 11 cells-09-01076-f011:**
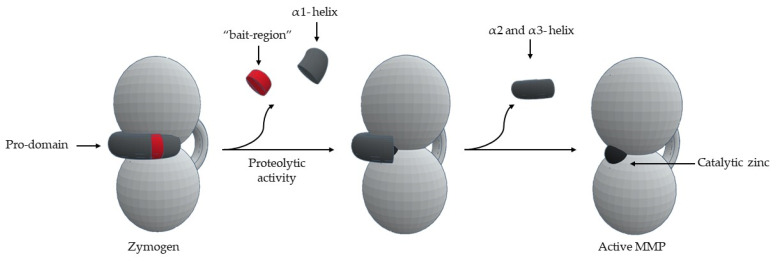
Schematic representation of MMP activation.

**Figure 12 cells-09-01076-f012:**
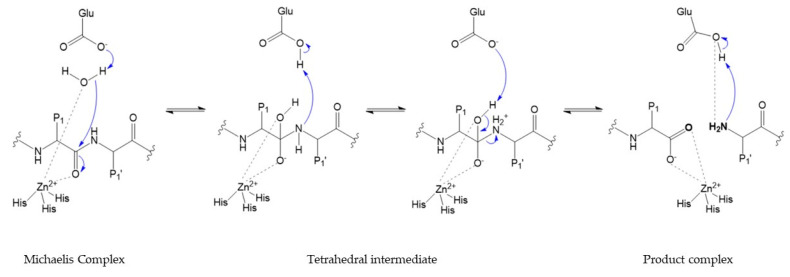
Catalytic mechanism.

**Table 1 cells-09-01076-t001:** Examples of diseases caused by deregulation of MMPs.

Pathologies	Diseases
Tissue destruction	Cancer invasion and metastasis
	Arthritis
	Ulcers
	Periodontal diseases
	Brain degenerative diseases
Fibroses	Liver cirrhosis
	Fibrotic lung disease
	Otosclerosis
	Atherosclerosis
	Multiple sclerosis
Weakening of matrix	Dilated cardiomyopathy
	Aortic aneurysm
	Varicose veins

**Table 2 cells-09-01076-t002:** Domain and presence in MMPs.

Domain	Presence
Signal Peptide	All MMPs
Pro-domain	All MMPs
Catalytic	All MMPs
Hemopexin-like	All MMPs, except in MMP-7, -23, and -26
Fibronectin	Only MMP-2 and -9
Vitronectin insert	Only MMP-21
Type I transmembrane	Only MMP-14, -15, -16, and -24
Cytoplasmic	Only MMP-14, -15, -16, and -24
GPI anchor	Only MMP-17 and -25
Cysteine Array Region	Only MMP-23
IgG-like domain	Only MMP-23

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
