# Peer review of "Structure and Function of Human Matrix Metalloproteinases"

_cells, 2020, doi:10.3390/cells9051076_

Round 1
Reviewer 1 Report
General comments:
>
> This review is original and seems very interessant. however I am not competent for the part on MMPs. I ve corrected first paragraph and there are a lot of inaccuracies, so I guess it's the same in the rest of the text. The other reviewers will say it.

Author Response
Dear Review 1
Thanks for the corrections and suggestions.
i) All questions regarding abbreviations are now addressed in the text and spelled out the first time that they are mentioned
ii) Line 29 suggestion incorporated in the text
iii) line 33 corrections included now in the text
iv) more reference were added.
Best regards
Reviewer 2 Report
Referee report for structure and Function of human matrix metalloproteinases
This review is a well written summary of our current knowledge in the matrix metalloproteinase field. It discusses their role in matrix-collagen turnover, their structure and functions and give good summaries for the types of MMPs known. Some omissions and errors have occured as detailed below.
What is disappointing is the lack of citations of original work in this review. For example although Gross and Lapiere are referenced in the text (p.3 line 74) the references cited have nothing to do with their original work, citing a review by Murphy & Nagase from 2008 and a paper by Mannello & Medda from 2012. This is only a single example but this is occurring throughout the manuscript and needs adressing.
Page 1 line 15 to 16 should read: This review shows the complexity of MMPs.
Page 3 line 71 to 72: The sentence in the manuscript does not make sense. In particular the part “by making connective tissue stroma, including small blood vessels that are more fluid “. The referee assumes, that the authors meant to say, that blood vessels become leaky or permeable.
Page 4:
Line 87: The references cited are not all written by Huang et al. This needs to be amended.
Line 88: colarectal should be colorectal
Table 1: Otosclerosis should be osteosclerosis
Page 6 line 156: MMP1 should read MMP11.
Line 162: in their pro-domain C-terminal should maybe read at the C-terminal end of their pro-domain.
Table 3-page 18:
For MMP8 serine protease inhibitors should be added as substrates.
For MMP13 in the other information box active needs to be replaced with activate.
MMP18/collagenase-4 does not have a human orthologue
Page 2 of the same table under MMP7: do the authors mean Fas-ligand? (FAZ)
Author Response
Dear Review 2
Thanks for the corrections and suggestions.
- The references were updated
- Page 1 line 15 to 16 suggestion incorporated in the text.
- Page 3 line 71 to 72: correction of the sentence included in the text.
- Page 4: the reference was updated (Ref 26)
- Line 88: spelling corrected
- Table 1: spelling corrected
- Page 6 line 156: corrected to MMP11.
- Line 162: correction of the sentence included in the text.
- Table 3-page 18: changes in the table were made according to indications
The MMP-18 and -22 were eliminated because theses MMPs are not humans.
Best regards
Reviewer 3 Report
I like at least few things about this review article. First, it really dedicated to the announced topic. Second, their description of MMPs structure is accurate and figures are original.
However, I believe that several improvements can be made.
The text needs to be formatted and edited.
For instance, when I read that "Huang et al related that MMP-9 represents a potential biomarker which is overexpressed in several types of tumors (colarectal carcinoma, breast, pancreatic, ovaria, cervical,.." (line 88)
I understand that Huang et al REPORTED ("suggested" or whatever) that ...
However, some readers may not understand it so well.
In many other cases, I see that square brackets are not separated from the text. For instance, I see two misformattings of this kind in lines 85-86:
Literally, they wrote: " MMPs have been recently recognized as biomarkers in several fields (diagnosis, monitoring and treatment efficacy)[17], since their overexpression in diseases conditions is specific and elevated[17]".
Instead, I would like it to be written in this way:
Literally, they wrote " MMPs have been recently recognized as biomarkers in several fields (diagnosis, monitoring and treatment efficacy) [17], since their overexpression in diseases conditions is specific and elevated [17]".
Space needs to be added before each square bracket that separates the respective reference from the text. Otherwise, the text is looking ugly.
My suggestion to the authors is quite simple. You wrote a good article and you are going to publish it in a highly respected journal. Please, do a favor to yourself - hire an editor that is going to make your article looking much better!
In the end, I would like to acknowledge that the authors might not cited all key papers on this topic, however, they studied all cited papers very well. If I might suggest them another one, I would probably chose a recent Nova Science publishers book on matrix metalloproteinases. It is a new one. They published it in 2019 and it contains lots of reasonable information for the readers.
https://novapublishers.com/shop/a-closer-look-at-metalloproteinases/
If you check it out, you may find reasonable to refer the readers to some papers in it.
After all, I would like to conclude that I want to see their paper published. However, I also suggest to the authors to make some improvements to the text. Thank you!
Author Response
Dear Review 3
Thanks for the corrections and suggestions.
i)The text is now more carefully formatted and edited.
ii)Line 88 sentence was corrected according to suggestion
iii) The square brackets are not separated from the text was a systematic error that was correct in all the text
iv)Line 91 text was modified according to sugestion
v) Reference updated (Ref.10)
best regards
Round 2
Reviewer 1 Report
Dear authors,
the text is so corrected that it is very difficult to verify
which corrections have been made or not.But ligne 23, in my previous report, I indicated that Tenascin was not a proteoglycan. This is not corrected.
So, without a clear version of the text, I can not accept it.
please accept my apologies
Author Response
Dear reviewer 1
We had to accommodate the comments corrections and suggestions of all 3 reviewers so some changes were necessary implemented .
The specific points mentioned were addressed and this last was corrected in the final text.
We hope that the manuscript is now in acceptable form, and we thank the pertinent points that improved our work.
with best regards
Jorge Caldeira